# A trafficome-wide RNAi screen reveals deployment of early and late secretory host proteins and the entire late endo-/lysosomal vesicle fusion machinery by intracellular *Salmonella*

**Alexander Kehl**[1,2¤]*, **Vera Göser**[1], **Tatjana Reuter**[1], **Viktoria Liss**[1], **Maximilian Franke**[1], **Christopher John**[1], **Christian P. Richter**[2], **Jörg Deiwick**[1], **Michael Hensel**[1,3]*

**1** Division of Microbiology, University of Osnabrück, Osnabrück, Germany, **2** Division of Biophysics, University of Osnabrück, Osnabrück, Germany, **3** CellNanOs–Center for Cellular Nanoanalytics, Fachbereich Biologie/Chemie, Universität Osnabrück, Osnabrück, Germany

¤ Current address: Institute of Hygiene, University of Münster, Münster, Germany
* alexander.kehl@ukmuenster.de (AK); Michael.Hensel@uni-osnabrueck.de (MH)

**Data Availability Statement:** All relevant data are within the manuscript and its Supporting Information files.

## Abstract

The intracellular lifestyle of *Salmonella enterica* is characterized by the formation of a replication-permissive membrane-bound niche, the *Salmonella*-containing vacuole (SCV). As a further consequence of the massive remodeling of the host cell endosomal system, intracellular *Salmonella* establish a unique network of various *Salmonella*-induced tubules (SIT). The bacterial repertoire of effector proteins required for the establishment for one type of these SIT, the *Salmonella*-induced filaments (SIF), is rather well-defined. However, the corresponding host cell proteins are still poorly understood. To identify host factors required for the formation of SIF, we performed a sub-genomic RNAi screen. The analyses comprised high-resolution live cell imaging to score effects on SIF induction, dynamics and morphology. The hits of our functional RNAi screen comprise: i) The late endo-/lysosomal SNARE (soluble *N*-ethylmaleimide-sensitive factor attachment protein receptor) complex, consisting of STX7, STX8, VTI1B, and VAMP7 or VAMP8, which is, in conjunction with RAB7 and the homotypic fusion and protein sorting (HOPS) tethering complex, a complete vesicle fusion machinery. ii) Novel interactions with the early secretory GTPases RAB1A and RAB1B, providing a potential link to coat protein complex I (COPI) vesicles and reinforcing recently identified ties to the endoplasmic reticulum. iii) New connections to the late secretory pathway and/or the recycling endosome via the GTPases RAB3A, RAB8A, and RAB8B and the SNAREs VAMP2, VAMP3, and VAMP4. iv) An unprecedented involvement of clathrin-coated structures. The resulting set of hits allowed us to characterize completely new host factor interactions, and to strengthen observations from several previous studies.

**Funding:** This work was supported by BMBF grant 0315834D 'Medizinische Infektionsgenomik', and the DFG by priority programme SPP 1580 grant HE 1964/18-2 to M.H, the Z project of SFB944. The funders had no role in study design, data collection and analysis, decision to publish, or preparation of the manuscript.

**Competing interests:** No authors have competing interests

## Author summary

The facultative intracellular pathogen *Salmonella enterica* serovar Typhimurium induces the reorganization of the endosomal system of mammalian host cells. This activity is dependent on translocated effector proteins of the pathogen. The host cell factors required for endosomal remodeling are only partially known. To identify such factors for the formation and dynamics of endosomal compartments in *Salmonella*-infected cells, we performed a live cell imaging-based RNAi screen to investigate the role of 496 mammalian proteins involved in cellular logistics. We identified that endosomal remodeling by intracellular *Salmonella* is dependent on host factors in the following functional classes: i) the late endo-/lysosomal SNARE (soluble *N*-ethylmaleimide-sensitive factor attachment protein receptor) complex, ii) the early secretory pathway, represented by regulator GTPases RAB1A and RAB1B, iii) the late secretory pathway and/or recycling endosomes represented by GTPases RAB3A, RAB8A, RAB8B, and the SNAREs VAMP2, VAMP3, and VAMP4, and iv) clathrin-coated structures. The identification of these new host factors provides further evidence for the complex manipulation of host cell transport functions by intracellular *Salmonella* and should enable detailed follow-up studies on the mechanisms involved.

## Introduction

The food-borne, facultative intracellular pathogen *Salmonella enterica* serovar Typhimurium (STM) is the etiological agent of gastroenteritis in humans or systemic infections in mice [1]. An early step in disease is the active invasion of epithelial cells. This process is dependent on the translocation of effector proteins by STM into the host cell through a type 3 secretion system (T3SS) encoded by genes in *Salmonella* pathogenicity island 1 (SPI1) [2, 3].

After invasion STM, similar to many other intracellular pathogens, establish a replicative niche in host cells, termed *Salmonella*-containing vacuole (SCV). This process is dependent on the function of a distinct T3SS, encoded by genes in SPI2 [4, 5] and translocating another set of effectors [6]. Though initially associating with markers of the early endosome (EE) such as EEA1 and the small GTPase RAB5 [7, 8], the SCV finally acquires several markers of the late endosome (LE). These markers include lysosome-associated membrane proteins (LAMPs) [9, 10], the vacuolar ATPase [11], and RAB7 [12, 13]. Concurrently, other canonical organelle markers such as the mannose-6-phosphate receptor are excluded [14].

A unique feature of STM among intravacuolar bacteria is the formation of a diverse array of long tubular structures, *Salmonella*-induced tubules (SIT) [15]. The first SIT discovered are the LAMP-decorated *Salmonella*-induced filaments (SIF) [16, 17]. Moreover, SIF have been structurally characterized, revealing the presence of a double membrane tubular network [18, 19]. The host-derived membranes forming SCV, SIF, and other tubular compartments are collectively termed *Salmonella*-modified membranes (SMM).

The repertoire of bacterial effector proteins necessary for the formation of SMM is quite well-characterized, with the SPI2-T3SS effector protein SifA being the most important factor [20, 21]. However, much less is known about corresponding host factors required for biogenesis of SMM. One crucial factor in SIF biogenesis is the SifA- and kinesin-interacting protein SKIP (a.k.a. PLEKHM2). In conjunction with the effectors SifA and PipB2 [22, 23] and the small GTPase ARL8B [24, 25], SKIP mediates kinesin-1 interaction and thus a link to the microtubule cytoskeleton and organelle motility [26].

Several attempts were made to analyze the interactions of STM with host factors in a systematic manner. These comprise RNA inference (RNAi) screens aiming at different parts of the STM infection process. Two genome-scale screens targeted the invasion [27, 28], while three screens focused on intracellular replication with two sub-genomic screens covering kinases and corresponding phosphatases, respectively [29, 30], and a genome-wide screen [31]. Additionally, two recent proteomic studies also shed light on interactions of intracellular STM with host cells. Vorwerk et al. [32] characterized the proteome of late SMM, while Santos et al. focused on early and maturing SCV [33].

All of these studies identified host factors yet unprecedented in STM pathobiology and showed the general value of such systematic approaches. However, none of these approaches targeted specifically SIF, thus a host-SIF interactome is far from complete. Therefore, we established a targeted RNAi screen comprising 496 human genes mostly involved in cellular logistics to identify host factors involved in the formation of SIF. Using stably LAMP1-GFP-transfected HeLa cells, we performed automated microscopy on a spinning disk confocal microscope (SDCM) system with time-lapse live cell imaging (LCI) of STM infection and scored for altered SIF formation as phenotypic readout. Investigating high-scoring hits of the RNAi screen, we validated several so far unknown host-SIF interactions by LCI: (i) involvement of the late endo-/lysosomal soluble *N*-ethylmaleimide-sensitive factor attachment protein receptor (SNARE) complex and its interaction partners, (ii) interactions of SIF with early secretory RAB1A/B, (iii) late secretory RAB3A, RAB8A/B, and VAMP2/3/4, and (iv) an interaction of SIF with clathrin-coated structures.

## Results

### RNAi screen setup and evaluation

We aimed to identify host cell factors that are required for the endosomal remodeling induced by intracellular STM in a SPI2-T3SS-dependent manner. For this, an RNAi screen was performed with siRNAs predominantly targeting mammalian genes involved in cellular logistics and trafficking (75.2% categorized in intracellular transport according to Gene Ontology [GO] terms). This subset of 496 genes was termed 'trafficome' and is listed in S1 Table (along with the GO terms from which these genes were selected). Such a screen necessitates specific considerations and controls with the major ones described below, and further experimental issues detailed in Suppl. Materials.

As a phenotypic readout for STM-induced endosomal remodeling, we specifically scored the formation of SIF in infected cells. SIF show a highly dynamic behavior in their early phase after formation, with constant elongation and retraction [34, 35]. Thus, in contrast to previous RNAi screens done by analyzing fixed cells, we decided to perform this screen by LCI to obtain maximal phenotypic information. A previously established HeLa cell line stably transfected with LAMP1-GFP as the marker for SIF [18] was used as the host cell.

As controls for STM-induced phenotypes, we used STM wild type (WT), capable in SIF induction, and an isogenic strain defective in SsaV, a central component of the SPI2-T3SS, and thus unable to induce SIF formation (Fig 1A). As a control for successful reverse transfection in general, we analyzed the lethal effect of an siRNA directed against polo-like kinase 1 (PLK1), a cell cycle control protein. The knockdown of this protein leads initially to a cell cycle arrest, and ultimately to cell death, as shown in Fig 1B. Besides, a phenotype-related control was established, i.e. a knockdown of a host factor already known to be essential for SIF formation. A host factor directly involved in SIF formation is SKIP [22]. This study already successfully used SKIP silencing, thus we used an siRNA with the same sequence as control. Real-time PCR indicated that the siRNA targeting SKIP yielded not a complete but significant

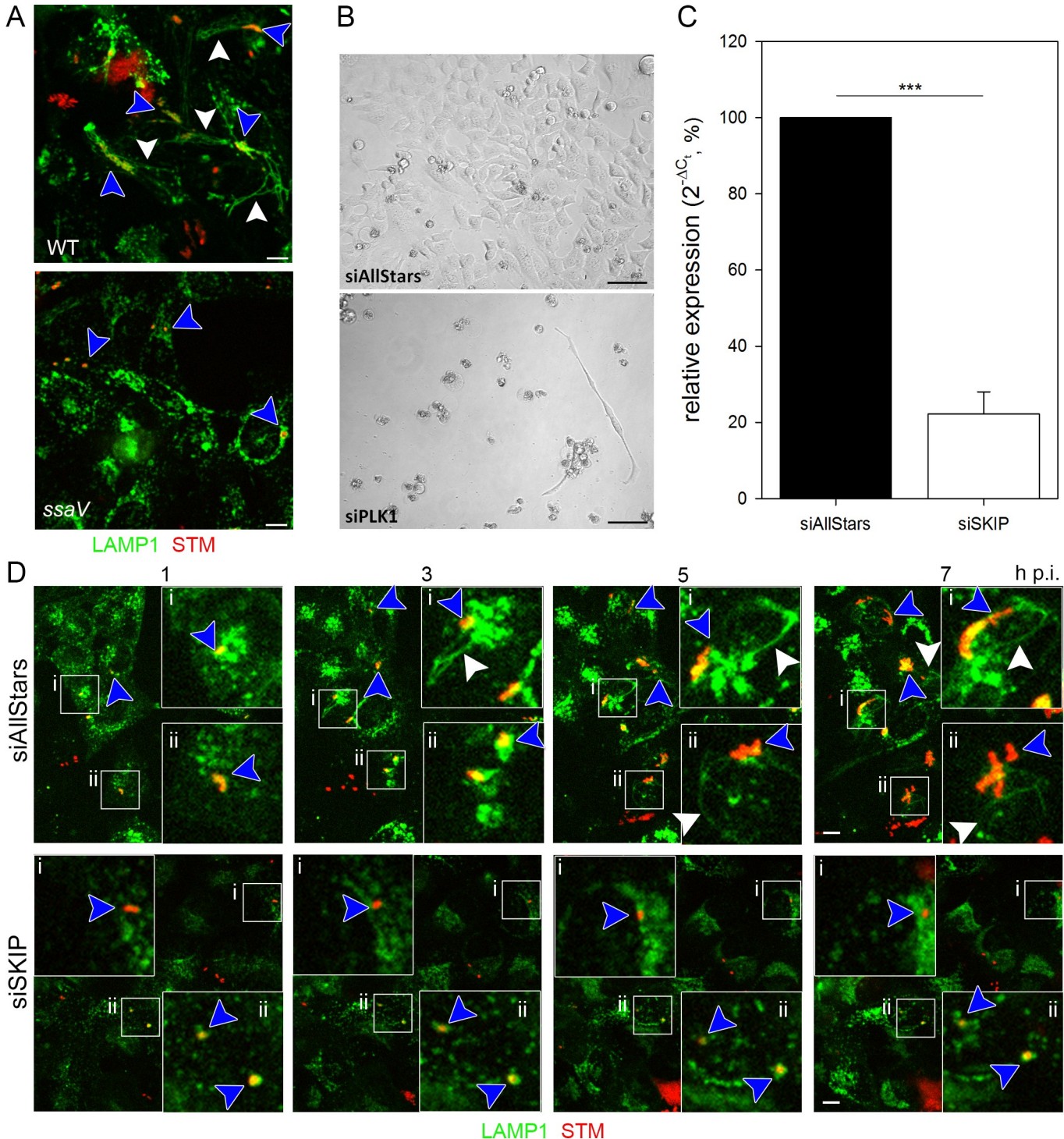

**Fig 1. RNAi screen setup and validation.** A) Intracellular phenotypes of STM under screening conditions. HeLa LAMP1-GFP cells were infected with mCherry-labelled STM WT or *ssaV* strains and imaged live 8 h p.i. by SDCM. Images display the presence of STM in LAMP1-positive SCV (blue arrowhead), the induction of SIF formation by STM WT (white arrowhead), and the lack of SIF formation by the STM *ssaV* strain. Scale bar, 10 μm. B) Controls for siRNA-mediated knockdown. HeLa LAMP1-GFP cells were reverse transfected with scrambled AllStars siRNA or PLK1 siRNA for 72 h and then imaged. Scale bar, 20 μm. C) Validation of SKIP siRNA knockdown. HeLa LAMP1-GFP cells were reverse transfected with AllStars or SKIP siRNA for 72 h. Then, RT-PCR targeting *SKIP* was performed. Depicted is the mean with standard deviation of three biological replicates (*n* = 3) each performed in triplicates. Statistical analysis was performed using Student's *t*-test and indicated as: ***, *p* < 0.001. D) SKIP knockdown as a control for the inhibition of SIF formation. HeLa LAMP1-GFP cells were first reverse transfected with AllStars or SKIP siRNA for 72 h. Then, cells were infected with mCherry-labelled STM WT (MOI = 15) and

imaged live by SDCM 1–7 h p.i in hourly intervals. Blue arrowheads indicate SIF-forming or non-SIF-forming single bacteria or microcolonies, white arrowheads indicate SIF. Scale bar, 10 μm.

knockdown with a reduction to ca. 22% (Fig 1C). Transfection with AllStars siRNA did not affect SIF formation and dynamics throughout the infection (Fig 1D, S1 Movie), while a SKIP knockdown abolished SIF formation (Fig 1D, S2 Movie) and reduced intracellular replication of STM. Though this outcome does not completely exclude off-target effects, the phenotypic/visual control showed at least the intended purpose of the SKIP siRNA being fulfilled. The partial knockdown explains the rare appearance of SIF. Taken together, the establishment of the proper controls allowed us the execution of a larger scale RNAi screen.

## The RNAi trafficome screen

The complete workflow of the RNAi screen executed is summarized in Fig 2. First, siRNAs (with three distinct siRNAs per target) were automatically spotted onto 24 96-well plates per biological replicate (with the total screen being performed in three biological replicates). Additionally, each plate contained siAllStars, siPLK1, and siSKIP as negative and positive siRNA controls, and as a phenotype-specific control, respectively. HeLa LAMP1-GFP cells were seeded onto siRNAs for reverse transfection, incubated for 72 h, and subsequently infected with mCherry-labelled STM WT or *ssaV*. The formation of SIF was followed by LCI acquiring eight positions per well with single bacteria-focused Z-planes with an SDCM from 1–7 h post infection (p.i.) with hourly intervals.

We set out to execute the analysis by visual inspection following the example of Stein et al. [20], who performed a mutant library screen to identify bacterial factors involved in SIF formation. As the dynamic nature of SIF and the phenotypic heterogeneity in the cellular context excluded fully automated analysis, we decided to perform the analysis by visual inspections and used a MATLAB-based tool named SifScreen to support data input and collection (Fig 2). This tool queried the presence of SIF in the examined field of view as the main feature in a binary manner (for detailed information see S1 Text). The scoring was always performed by analyzing the complete time-lapse movies for each position allowing to identify SIF formation due to its dynamic nature more easily.

Since siRNA silencing usually does not yield 100% loss of function, we did not expect a complete lack of SIF in each of the eight images per well. Furthermore, since a considerable number of cells were present per image (roughly 10–30 cells, depending on applied siRNA and position on plate) a single SIF-forming cell would have prompted a SIF-positive scoring, even if generally a SIF-negative knockdown might have occurred. Thus, we decided to define an overall SIF-abolishing hit with a comparably high cutoff of 50%, i.e. if less than 50% of the images showed SIF. However, this cutoff did not take into consideration knockdowns possibly affecting cell viability in general or other circumstances compromising the analysis. Because parameters such as host cell viability were also queried by SifScreen, this assessment allowed us to differentiate between 'true hits' and 'possible hits,' scoring the former and latter with values of 3 and 1, respectively. This is important as other influences, such as a decreased invasion, could affect the outcome of the scoring, even though they might not have been readily detectable. Additionally, to avoid a possible bias due to visual analysis, each screening plate was analyzed independently by two investigators. With the screen performed in biological triplicates ($n = 3$), we subsequently compiled all scoring data for each host target from both investigators, also pooling the results of the three individual siRNAs per target. This summary resulted in a list of final hits shown in S2 Table, in which hits with a cumulative scoring of 1–4, 5–7, or ≥8

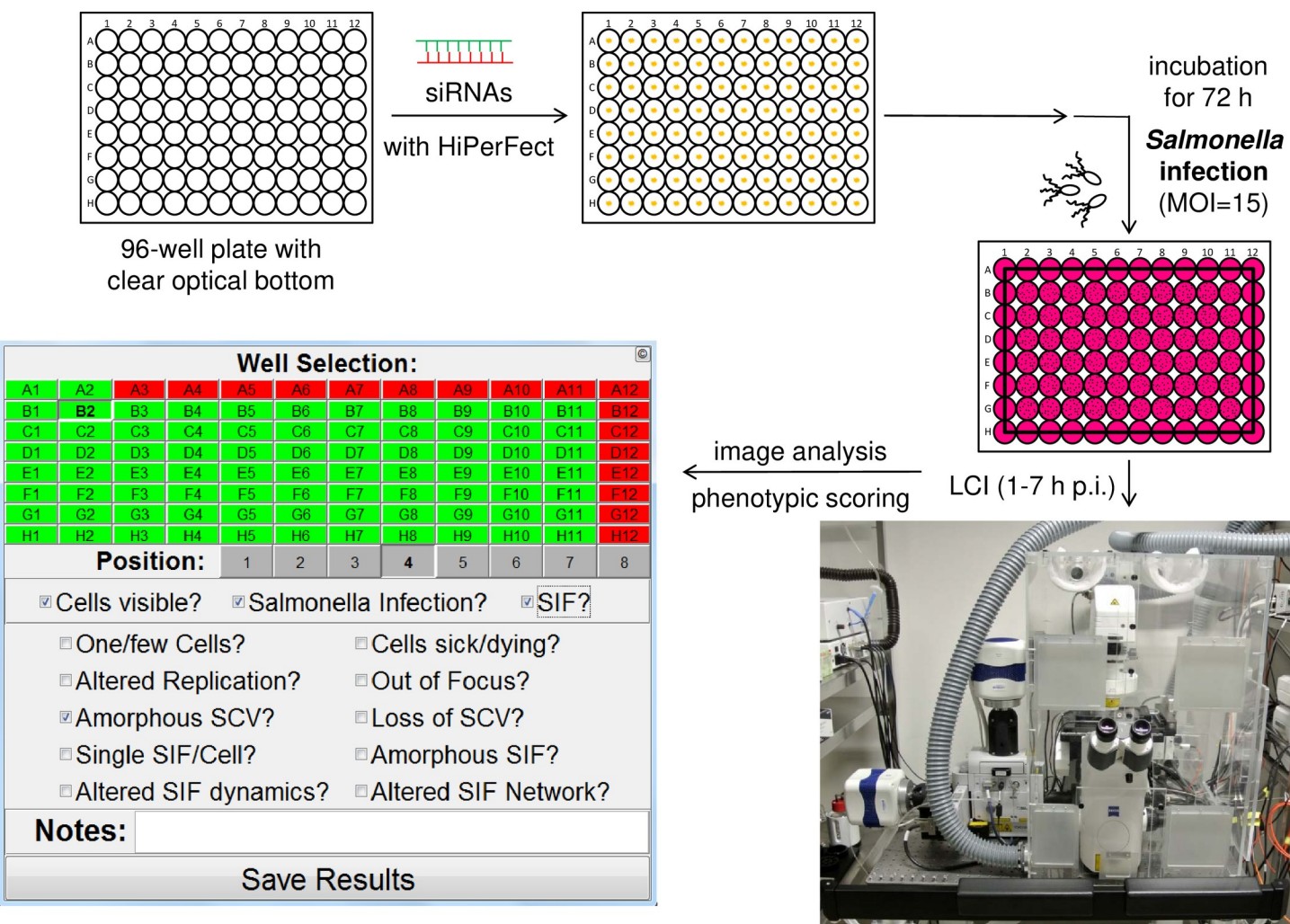

**Fig 2. Basic workflow of the trafficome RNAi screen.** 24 96-well plates with clear optical bottoms per biological replicate (with 3 biological replicates in total, *n* = 3) were automatically spotted with siRNAs. HeLa LAMP1-GFP cells were seeded for reverse transfection with each plate also containing negative, positive, and phenotype-specific controls. After 72 h of incubation, infection with STM WT (and *ssaV* as control, MOI = 15) was performed, followed by LCI for the acquisition of eight positions per well with single bacteria-focused Z-planes with hourly intervals of imaging from 1–7 h p.i. on an SDCM system. Subsequent phenotypic scoring was performed using the SifScreen utility. The MATLAB-based data input mask allows the entry of well- and position-specific information on general cell behavior and *Salmonella*/SMM phenotypes and the generation of a results report.

were classified as low-, mid- and high-ranking hits, respectively. Examples of time-lapse acquisitions of selected siRNAs are given in S1 Fig.

Approximately 81% (404 of 496) of the trafficome targets scored to varying degrees positive, underlining the general importance of trafficking processes for SIF formation. Table 1 shows selected high-ranking hits involved in trafficking and cytoskeleton biology. These hits clearly show the involvement of all protein classes necessary for the vesicle budding and fusion machinery, the core of cellular trafficking. These comprise: (i) small GTPases, especially Rab GTPases, as primary regulators [36–38]; (ii) vesicle coats and their adaptors as cargo and budding mediators [39–43]; (iii) cytoskeleton components as the basis for vesicle motility [44]; (iv) tethering factors as part of the fusion specification [45, 46]; (v) SNAREs as the primary fusion agents [45, 47, 48]. Besides, this list includes hits of diverse subcellular origin, encompassing the complete secretory and endo-/lysosomal system, i.e. endoplasmic reticulum (ER), Golgi

**Table 1. High-ranking trafficome hits (scoring cutoff of ≥8; see main text for scoring details) involved in trafficking and cytoskeleton biology (see S1 Table for more details on individual host factors).**

| Gene symbol | Full name[1] | Localization[2] |
|---|---|---|
| **Small GTPases and interacting proteins** | | |
| *Arf family* | | |
| ARL1 | ADP-ribosylation factor-like GTPase 1 | Golgi |
| *Rab family* | | |
| RAB1A | RAB1A, member RAS oncogene family | ER, Golgi, EE |
| RAB11A | RAB11A, member RAS oncogene family | RE, vesicle, PM |
| *Ras family* | | |
| RHOB | ras homolog family member B | nucleus, LE, PM |
| RHOT1 | ras homolog family member T1 | mitochondrion OM |
| *Interacting proteins* | | |
| G3BP2 | G3BP stress granule assembly factor 2 | cytoplasm |
| RAB3GAP2 | RAB3 GTPase activating non-catalytic protein subunit 2 | cytoplasm |
| **Vesicle coats and adaptors** | | |
| *BBSome* | | |
| BBS4 | Bardet-Biedl syndrome 4 | CS/MTOC |
| *Clathrin coats* | | |
| AGFG1 | ArfGAP with FG repeats 1 | nucleus, vesicle |
| AP2A1 | adaptor related protein complex 2 subunit alpha 1 | CCV, PM |
| AP3D1 | adaptor-related protein complex 3 subunit delta 1 | CCV, Golgi |
| CLTA/B | clathrin light chain A and B | CCV |
| CLTC | clathrin heavy chain | CCV |
| GGA3 | Golgi-associated, gamma adaptin ear containing, ARF binding protein 3 | Golgi, endosome |
| SYNRG | synergin gamma | Golgi |
| *COP-I* | | |
| ARCN1 | archain 1 | Golgi, vesicle |
| COPA/B1/B2/G1 | COPI coat complex subunit alpha, beta 1, beta 2, and gamma | Golgi, vesicle |
| TMED10 | transmembrane p24 trafficking protein 10 | ER, Golgi, vesicle |
| *COP-II* | | |
| SEC24D | SEC24 homolog D, COPII coat complex component | ER, Golgi, vesicle |
| *Retromer* | | |
| VPS35 | VPS35 retromer complex component | EE, LE |
| **Tethering factors** | | |
| *Exocyst complex* | | |
| EXOC5 | exocyst complex component 5 | cytoplasm |
| *TRAPPIII complex* | | |
| TRAPPC8 | trafficking protein particle complex 8 | Golgi |
| **SNAREs** | | |
| *Qa-SNAREs* | | |
| STX5 | syntaxin 5 | Golgi |
| STX7 | syntaxin 7 | EE, LE |
| *Qb,c-SNAREs* | | |
| SNAP23 | synaptosome associated protein 23 | PM |
| *R-SNAREs* | | |
| SEC22B | SEC22 homolog B, vesicle trafficking protein (gene/pseudogene) | ER, Golgi |
| VAMP7 | vesicle associated membrane protein 7 | ER, Golgi, LE/ lysosome, vesicle, PM |
| *Interacting proteins* | | |

(*Continued*)

**Table 1.** (Continued)

| Gene symbol | Full name[1] | Localization[2] |
|---|---|---|
| NAPA | NSF attachment protein alpha | membrane |
| STXBP2 | syntaxin binding protein 2 | PM |
| **Cytoskeleton and motor proteins** | | |
| *Kinesins* | | |
| KIF1A/B/C | kinesin family member 1A, B, and C | CS |
| *Dyneins* | | |
| DYNC1H1 | dynein cytoplasmic 1 heavy chain 1 | CS |
| *Microtubule-associated proteins* | | |
| CEP57 | centrosomal protein 57 | CS/MTOC, nucleus |
| MAP1A | microtubule associated protein 1A | CS |
| *Myosins* | | |
| MYH10 | myosin heavy chain 10 | CS |
| *Actin filament membrane linkers* | | |
| ANK3 | ankyrin 3 | CS |
| FLNA | filamin A | CS |
| **ESCRT complexes** | | |
| *Adaptors* | | |
| HGS | hepatocyte growth factor-regulated tyrosine kinase substrate | EE, MVB |
| *AAA ATPase* | | |
| VPS4A/B | vacuolar protein sorting 4 homolog A and B | MVB |
| **Miscellaneous** | | |
| ERGIC1 | endoplasmic reticulum-golgi intermediate compartment 1 | ER, Golgi |
| SNX15 | sorting nexin 15 | vesicle |
| SORT1 | sortilin 1 | nucleus, ER, Golgi, endo-/lysosome |
| VCP | valosin containing protein | nucleus, ER |

[1] according to NCBI Gene

[2] subcellular localization according to UniProt; CCV = clathrin-coated vesicle, CS = cytoskeleton, EE = early endosome, ER = endoplasmic reticulum, LE = late endosome, MTOC = microtubule-organizing center, MVB = multivesicular body, OM = outer membrane, PM = plasma membrane, RE = recycling endosome

apparatus, endo-/lysosomes. Supporting these allocations, the interaction network of the hits from Table 1 shows several distinct clusters (Fig 3). Two of the clusters are connected to cyto-skeleton biology (also interconnected if lower-ranking hits are included). Another cluster is SNARE-centered, including RAB1A, with RAB11A as a node between this cluster and one of the cytoskeleton-related clusters. Lastly, one cluster is associated with COPI and clathrin-coated vesicles (CCVs). Collectively, the overall results of the trafficome screen confirm the general importance of host trafficking factors in SIF biogenesis, indicating a crucial role for a plethora of as yet unprecedented factors in STM pathobiology.

## Validation of selected hits

To test the validity of our approach and the resulting hits, we focused on a subset of genes due to their presence in the noticeable interaction clusters depicted in Fig 3, or prior reports on involvement in STM pathobiology. HGS is part of the 'endosomal sorting complex required for transport' (ESCRT) complex ESCRT-0. Interaction of SPI1-T3SS effector SopB with HGS was previously reported [49], and HGS was one of the highest-ranking hits (S2 Table). Fur-thermore, RAB1A and RAB11A were included as highest-ranking Rab GTPases, with RAB11A

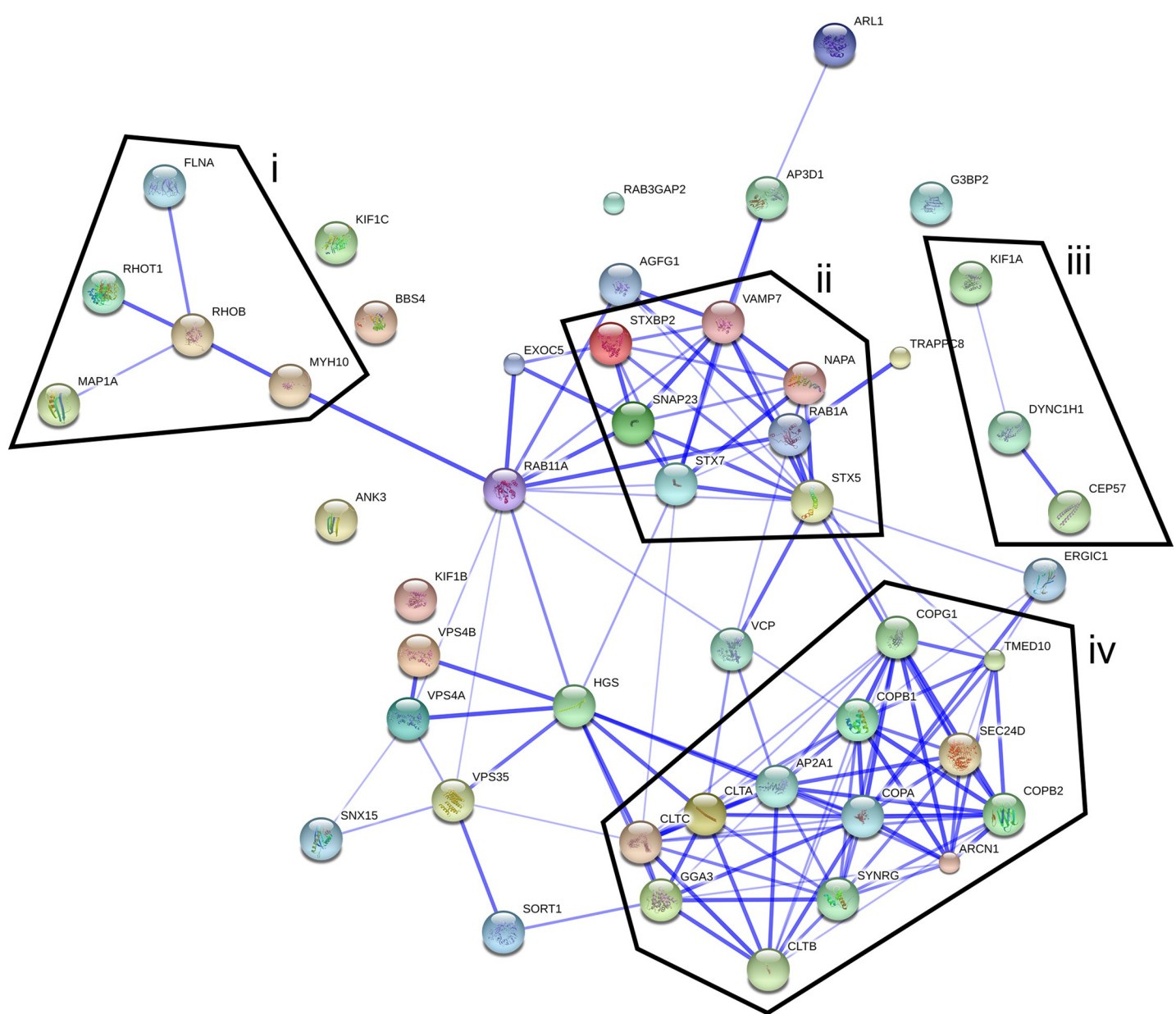

**Fig 3. Interaction network of selected trafficome hits.** The interaction of selected high-ranking hits (scoring cutoff of ≥8, see also Table 1) was visualized using the STRING database (confidence view). Borders delineate clusters related to the cytoskeleton (i, iii), SNAREs (ii), or COPI and clathrin-coated vesicles (iv).

previously being shown to colocalize with SCV as well as SIF [32, 50]. Consistently, RAB7A served as another well-established SCV- and SIF-localizing control [12, 50–52]. STX5, STX7, VAMP7, and VAMP8 were chosen due to being the highest-ranking SNAREs (except VAMP8 lacking from the trafficome), the previously shown colocalization of STX7 with SIF [50], and the recently reported essential role of VAMP7 in SIF biogenesis [33]. The AAA+ (ATPases associated with diverse cellular activities) protein VCP was included as another of the highest-ranking trafficking-related hits and another host factor already known to be important for proper SCV and SIF biogenesis via the STM effector SptP [53]. Finally, the VPS11 core component of the class C core vacuole/endosome tethering (CORVET) / homotypic fusion and protein sorting (HOPS) group of multisubunit tethering complexes (MTCs) was chosen due to

the HOPS complex functionally bridging RAB7 with late endo-/lysosomal SNAREs and the recent recognition of its essential role in STM replication and SCV and SIF biogenesis [54, 55].

The success of the silencing with individually ordered siRNAs for subsequent SIF quantification was first confirmed by RT-PCR with a consistently significant decrease in mRNA in most cases down to 5–10% compared to control siAllStars (S2A Fig, siSKIP served as the screen-inherent phenotype-specific control). As an mRNA reduction does not necessarily affect the actual protein levels due to potential long protein half-lives, we additionally tested the presence of selected targets, i.e. SKIP, RAB7A, and VPS11, by Western blot analysis (S2B and S2C Fig). We observed reduced protein levels for all targets, though the decrease for SKIP and VPS11 was moderate with 82.0% and 69.9%, respectively, compared to siAllstars-treated cells, whereas amounts of RAB7A decreased to 13.8%. Next, we determined the effect of silencing of the selected targets on SIF formation (Fig 4). The *ssaV* mutant strain and the knockdown with siSKIP served as screen-inherent SIF-abolishing controls. All knockdowns resulted in decreased SIF formation with the reduction being statistically significant except for siHGS and siSTX5. Regarding SKIP and VPS11, this decline also exhibits that moderate silencing on the protein level suffices to exert a biological effect. The siRAB7A had the highest impact that corresponds with the strong reduction of the RAB7A protein level, siVCP was second-highest, and the others ranged similar. Thus, the knockdowns of VAMP7 and VCP meet previous data (even though SIF abolishment regarding VCP depletion here is less pronounced) [33, 53].

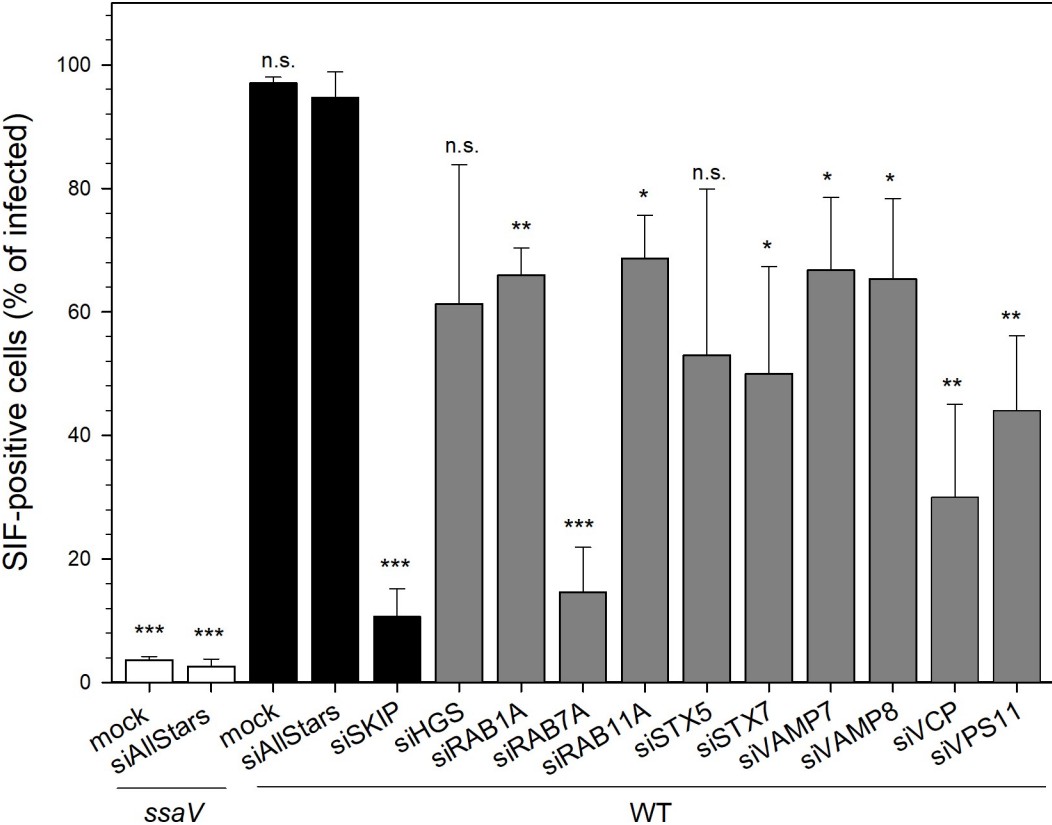

**Fig 4. Influence of host factor silencing on SIF formation.** HeLa LAMP1-GFP cells were not transfected (mock), or reverse transfected with siAllStars or the indicated siRNA, infected with STM WT or SPI2-deficient *ssaV* expressing mCherry as indicated, and SIF-positive infected cells were counted. Depicted are means with standard deviation for three biological replicates ($n = 3$). Statistical analysis was performed against siAllstars + WT with Student's *t*-test and indicated as: n.s., not significant; *, $p < 0.05$; **, $p < 0.01$; ***, $p < 0.001$.

Altogether, the effect of silencing various host targets on SIF formation demonstrates that our approach confirms known host factors, but also allows the identification of novel factors to be crucial for SIF biology.

## STM deploys membranes of early and late secretory, late endo-/lysosomal, and clathrin-coated origin in SIF biogenesis

The fact that host factors appear as hits in our screen clearly indicates a physiologically relevant role in SIF biogenesis. However, whether this role is by direct interaction or an indirect one involving several intermittent steps, remains unclear. Thus, we decided to analyze the localization of selected hits with regard to SIF (Fig 5, Fig 6, Fig 7). Even though mere colocalization in light microscopy is no ultimate proof of direct interaction, it is a first approximation as it potentially allows such a possibility.

For analyses of RAB GTPases (Fig 5), we again used RAB7A and RAB9A as positive controls, both showing a clear colocalization with SIF. Of the several Rab GTPases included in the trafficome RAB1A showed the highest score (S2 Table). RAB1 GTPases are responsible for anterograde ER-Golgi trafficking [56–59]. Importantly, RAB1A can be functionally substituted by RAB1B [60, 61] and an STM replication-targeted RNAi screen identified specifically RAB1B as a hit [31]. Hence, we analyzed the infection-related localization of both, RAB1A and RAB1B, and detected a partial and a strong colocalization of RAB1A and RAB1B, respectively, with SIF (Fig 5).

Another high-ranking hit with relation to RAB proteins was RAB3GAP2 (S2 Table), the non-catalytic subunit of the RAB3 inactivating GTPase-activating protein (GAP) complex [62]. RAB3 possesses four isoforms in mammals [63] and is involved in regulated exocytosis [64]. As neither the catalytic GAP subunit, RAB3GAP1, nor one of the four isoforms of RAB3 were present in the trafficome, we decided to analyze the localization of RAB3A and found a partial colocalization with SIF (Fig 5).

Besides, a mid-ranking hit was RAB8A (S2 Table), a Golgi- and endosome-localized RAB likewise involved in exocytic processes [65]. Interestingly, RAB8A isoform RAB8B was observed to be excluded from maturing SCVs ($\leq$ 3 h p.i.) [50]. Therefore, we analyzed the localization of both, RAB8A and RAB8B, and strikingly found a strong colocalization of not only RAB8A but also RAB8B with SIF (Fig 5).

As several RAB proteins participating in the late secretory system/exocytosis seem to play a role in SIF biogenesis, we additionally analyzed three SNAREs with exocytic roles not present in the trafficome: VAMP2, VAMP3, and VAMP4 [66–68], with VAMP2 also shown to be present on early SCV [69]. Apart from that, the presence of the two high-ranking SNARE hits STX7 and VAMP7 (S2 Table) on SIF was previously shown [33, 50]. However, SNAREs, are part of complexes of usually four proteins participating in membrane fusion and consisting of a single v-SNARE (on the vesicle or incoming membrane) and a ternary t-SNARE subcomplex (on the target or accepting membrane). VAMP7 is the v-SNARE in the SNARE complex for heterotypic LE/lysosome fusions with the t-SNAREs STX7, STX8, and VTI1B [70, 71] being replaced by VAMP8 in homotypic LE fusions [72, 73]. In fact, the presence of VTI1B and STX8 on early SCV [69, 74] and their role in STM replication [54], as well as the involvement of VAMP8 in STM invasion were already shown [75]. However, the interaction of VTI1B, STX8, and VAMP8 with SIF remains unclear, except VAMP8 silencing causing SIF reduction identified here (Fig 4). Thus, we analyzed the localization of VTI1B, STX8, and VAMP8 using STX7 and VAMP7 as controls. As shown in Fig 6, we detected a prominent association of STX7 and VAMP7 with SIF, and of VAMP2 and VAMP8. Colocalization of VTIB, STX8, VAMP3, and VAMP4 with SIF was also observed, however, these SNARE subunits showed a more heterogeneous distribution and only a fraction of SIF was positive for these candidates.

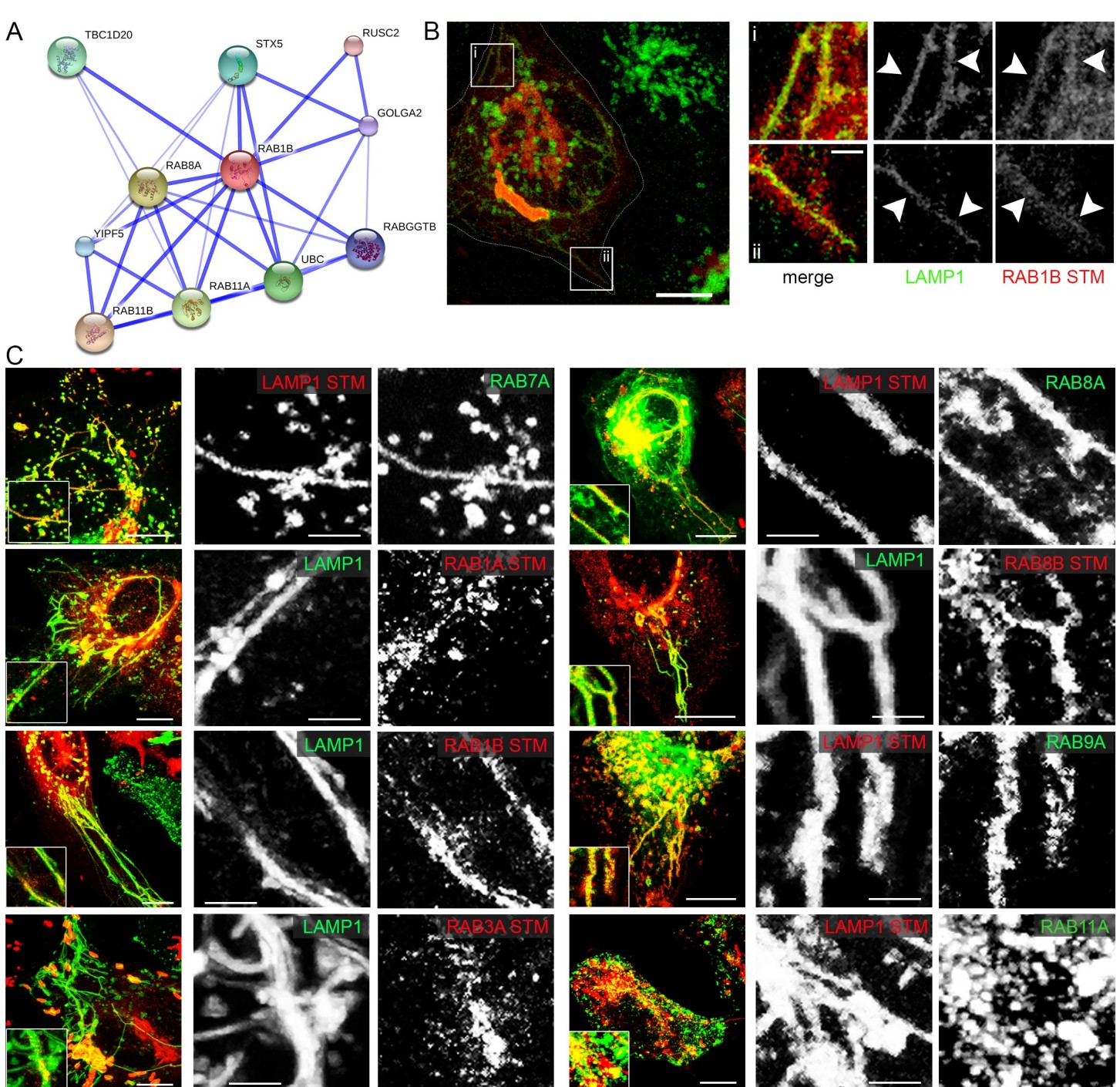

**Fig 5. RAB proteins identified by the trafficome screen colocalize with SIF and SCV.** A) Direct interaction network of RAB1B as visualized by STRING. B) and C) HeLa cells either stably transfected with LAMP1-GFP (green) or transiently transfected with LAMP1-mCherry (red) were co-transfected with plasmids encoding various RAB GTPases (RAB7A, RAB1A, RAB1B, RAB3A, RAB8A, RAB8B, RAB9A, RAB11A) fused to GFP (green) or mRuby2 (red) and then infected with STM WT expressing mCherry or GFP. Living cells were imaged from 6–9 h p.i. by CLSM and images are shown as maximum intensity projections (MIP). Insets magnify structures of interest and white arrowheads indicate colocalization with SIF. Scale bars, 10 μm (overviews), 1 μm (details).

Notably, both recent proteomic studies [32, 33] identified AP2A1 as being present on SMM/SCV, and this screen determined AP2A1 as high-ranking hit of host factors involved in endosomal remodeling (S2 Table). AP2A1 represents one of the two core subunit isoforms of

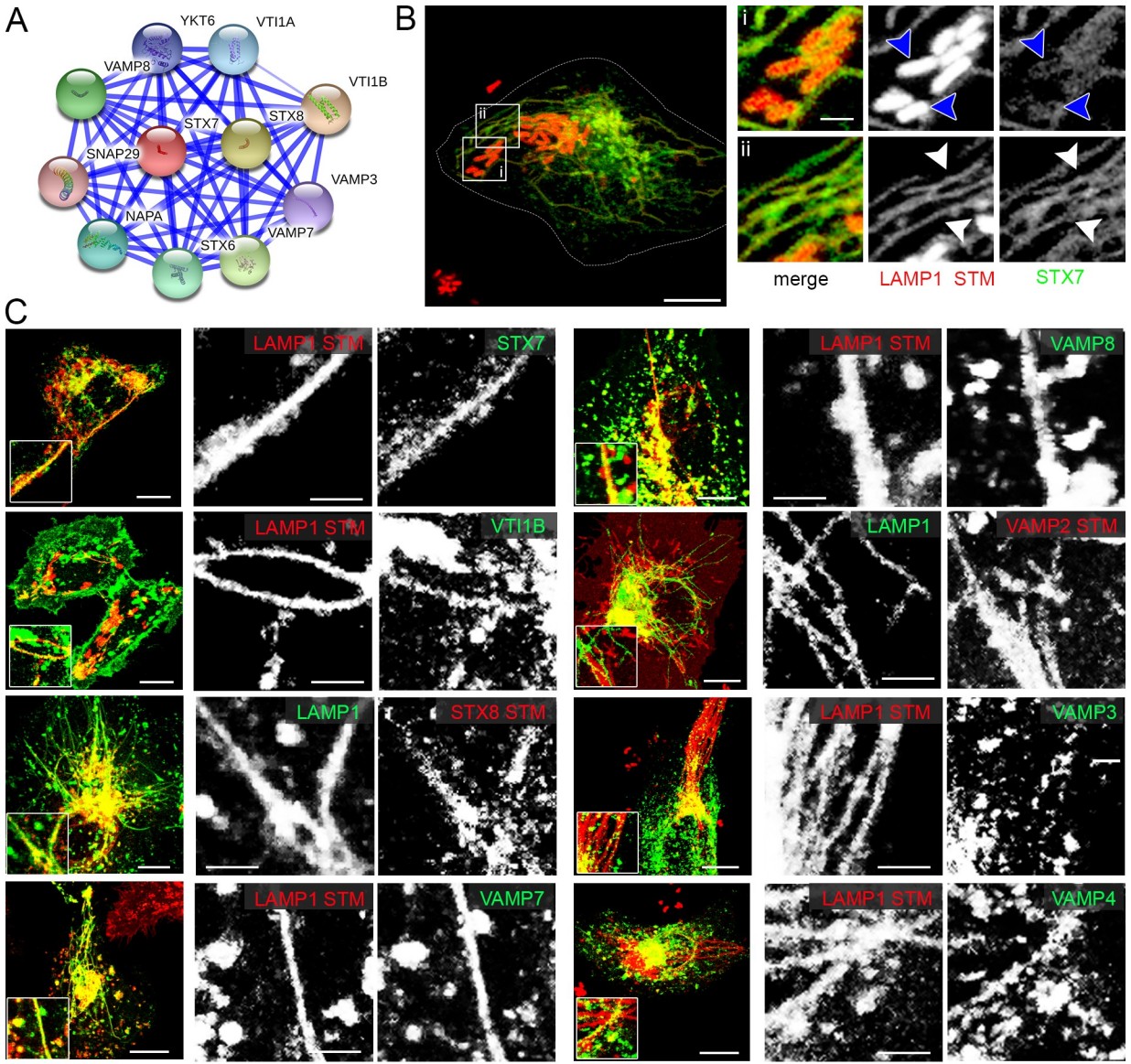

**Fig 6. SNARE proteins identified by the trafficome screen colocalize with SIF and SCV.** A) Direct interaction network of STX7 as visualized by STRING. B) and C) HeLa cells either stably transfected with LAMP1-GFP (green), or transiently transfected with LAMP1-mCherry (red) were co-transfected with plasmids encoding various SNAREs (STX7, VTIB, STX8, VAMP7, VAMP8, VAMP2, VAMP3, VAMP4) fused to GFP (green) or mRuby2 (red). Infection and imaging were performed as for Fig 5. Insets magnify structures of interest and white and blue arrowheads indicate colocalization with SIF and SCV, respectively. Scale bars, 10 μm (overviews), 1 μm (details).

the canonical AP-2 adaptor complex usually acting in clathrin-mediated endocytosis (CME) at the plasma membrane [76, 77]. With both clathrin light chains CLTA and CLTB, and the conventional heavy chain CLTC, the main coat components of CCV formation were among the high-scoring hits (S2 Table). We analyzed the localization of CLTA and observed a partial colocalization with SIF (Fig 7).

In conclusion, the colocalization of various host factors involved in cellular transport with SIF validated the results of the RNAi screen. These proteins are components of SIF tubules and, to a variable extent, required for the formation of SIF.

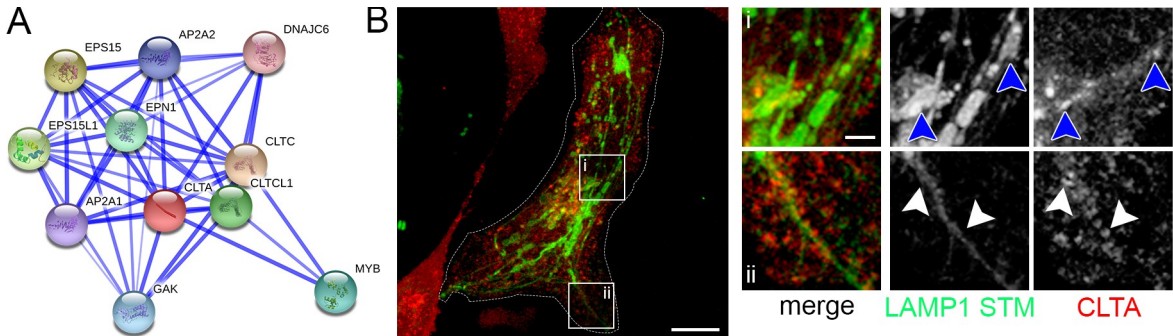

**Fig 7. CLTA identified by the trafficome screen colocalizes with SIF and SCV.** A) Direct interaction network of CLTA as visualized by STRING. B) HeLa cells stably transfected with LAMP1-GFP (green) were co-transfected with a plasmid encoding CLTA fused to mRuby2 (red). Infection and imaging were performed as for Fig 5. Insets magnify structures of interest and white and blue arrowheads indicate colocalization with SIF and SCV, respectively. Scale bars, 10 μm (overviews), 1 μm (details).

## Discussion

By applying a targeted RNAi screen, we identified several new host factors required for the formation of SIF and partially characterized interactions of host proteins with SMM. Our data strengthen the involvement of the late endo-/lysosomal SNARE complex, and reveal new interactions of SIF with RAB1, RAB3, and RAB8 GTPases, exocytic SNAREs, and clathrin-coated structures. The implications of these findings as discussed below are depicted in Fig 8. Several trafficome targets identified here as high-ranking hits besides those mentioned above were previously shown to be not only involved in infection biology in STM in general, but specifically in SCV and/or SIF biogenesis including: dynein–DYNC1H1 [78–80], filamin–FLNA [81], myosin II–MYH10 [82], VPS4A/B [49]. This also holds true for several mid-ranking hits: kinesin-1 –KIF5A/B [22–25, 83], PIKFYVE [84], RAB9A [85, 86], RAB14 [85], SCAMP3 [87].

Data complementary to our screen were recently provided by two proteomic studies. Our group analyzed the SMM proteome in the late phase of infection (8 h p.i.) [32] that contained several host proteins that are mid- or high-ranking hits in this screen (summarized in Table 2, first column). The colocalization of several of these proteins with SMM was shown by immunostaining or LCI in that study. Santos et al. [33] determined the proteomes of early and maturing SCV (30 min p.i. and 3 h p.i., respectively) again identifying proteins appearing as hits in this screen (see Table 2, second and third column). Taken together, these data strongly validate the approach deployed here.

The approach reported here has a major advantage compared to studies based on organelle proteomics [32, 33]. Proteomic analyses lead to the identification of the presence or absence of host factors on the organelle of interest, but a particular role in the biogenesis of this organelle cannot be implied directly [32, 33]. In our RNAi approach potentially each, or at least each high-ranking hit, points to a role in STM-induced endosomal remodeling. However, the functional role revealed by RNAi does not necessarily depend on localization of the host factor at SIF and/or SCV. The effect on endosomal remodeling may be mediated indirectly, involving several interacting partners. We analyzed the localization of selected host factors (Fig 5, Fig 6, Fig 7) and found several differences in the host factor sets identified by proteomics or by our approach. Nevertheless, there is a considerable overlap of host factors identified by both approaches, proteomics and RNAi, as represented in Table 2.

The presence on SIF and/or importance for SIF formation of RAB7, the HOPS complex, STX7, and VAMP7, as well as the direct fusion of late endo-/lysosomal-like VAMP7-positive vesicles with the SCV, was shown before [33, 50, 54]. These interactions indicate the

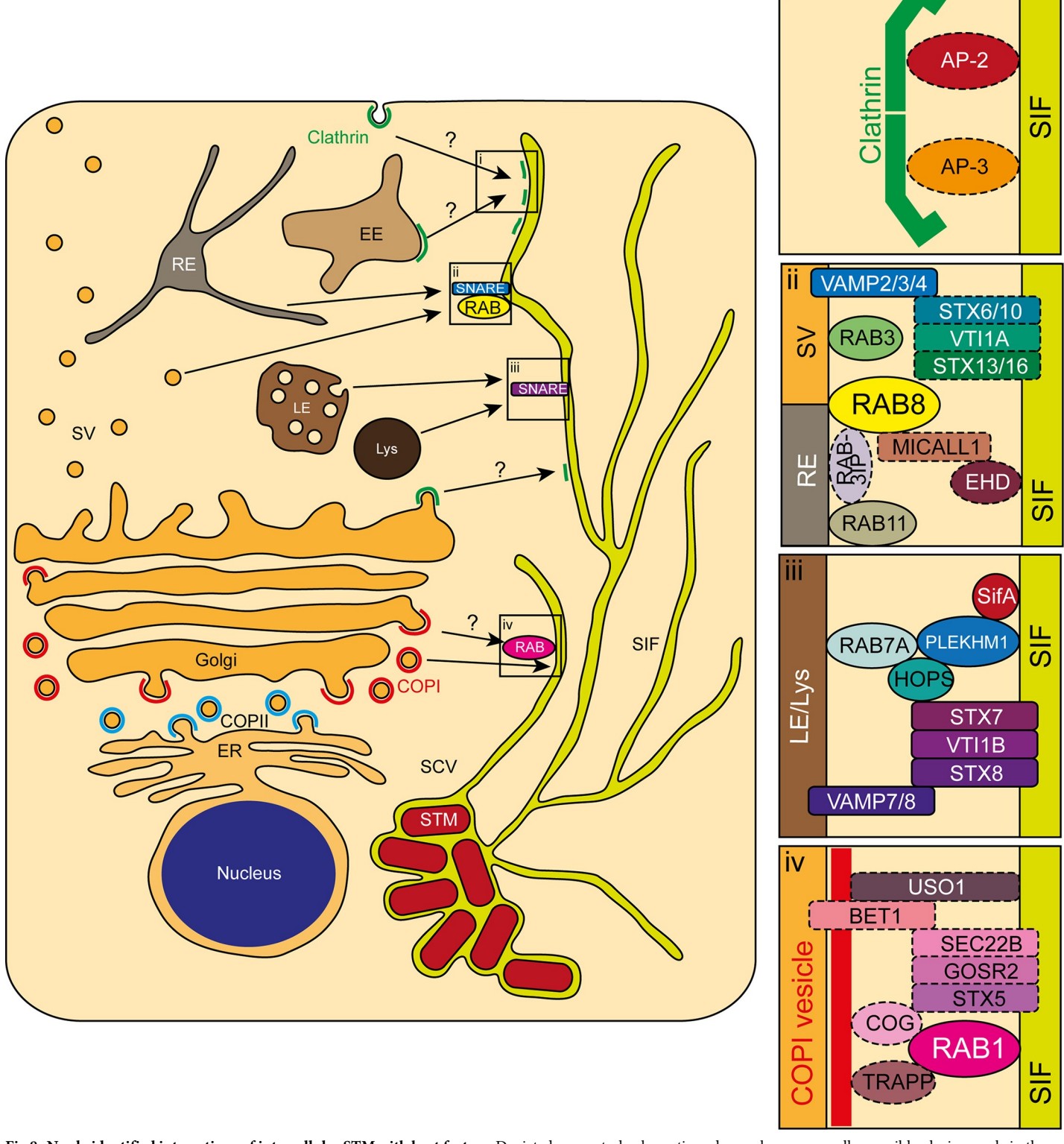

**Fig 8. Newly identified interactions of intracellular STM with host factors.** Depicted are central eukaryotic endomembrane organelles possibly playing a role in the newly identified interplays of host factors with SIF. Magnifications show the interactions of clathrin (i), late secretory and/or recycling-related RAB3A, RAB8A/B, and VAMP2/3/4 (ii), late endo-/lysosomal VTI1B, STX8, and VAMP8 (iii), and early secretory RAB1A/B (iv) with other host factors added as discussed in the text. Solid lines represent interactions identified here or otherwise known, dashed lines represent putative interactions. COP, coat protein complex; EE, early endosome; ER,

endoplasmic reticulum; LE, late endosome; Lys, lysosome; RE, recycling endosome; SCV, *Salmonella*-containing vacuole; SIF, *Salmonella*-induced filaments; STM, *S.* Typhimurium; SV, secretory vesicle.

involvement of the complete canonical mammalian late endo-/lysosomal vesicle fusion machinery in SIF biogenesis. Whether this interaction cascade also employs the canonical STX7, VTI1B, and STX8 was not fully clarified. Here, we expand this cascade by showing the physiological relevance of STX7 for SIF formation (Fig 4), and the presence of VTI1B and STX8 on SIF (Fig 6) as depicted in Fig 8iii. This cascade is possibly expanded by the host protein PLEKHM1, as the recruitment of RAB7 and the HOPS complex by SifA via the host protein PLEKHM1 and its involvement in SCV biogenesis was recently revealed [55], most likely also being involved in SIF biogenesis. Taken together, SifA seems to recruit the complete late endo-/lysosomal fusion machinery. Thus, SifA performs a dual role besides the binding of SKIP and the SIF mobility connected with it. This is also corroborated by the identification of interactions of SifA with STX7 and VAMP7 by a recent BioID screen [88]. Alternatively or in addition, SopD2 might be likewise involved as it was also shown to interact with STX7 and VAMP7 besides VTI1B in the same study.

**Table 2. Host proteins (gene symbols) identified as hits in the trafficome screen here that are also part of at least one distinct SMM proteome as identified in other studies (see S1 Table for more details on individual host factors).**

| 8 h p.i. SMM[1] | 30 min p.i. SCV[2] | 3 h p.i. SCV[2] |
|---|---|---|
| AP2A1 (AP-2) | - | AP2A1 |
| - | - | BET1 (SNARE) |
| - | CLTC (clathrin) | - |
| COPA/G1 (COPI)[3] | - | - |
| DYNC1H1 (dynein) | DYNC1H1 | - |
| - | ERGIC1 | ERGIC1 |
| - | ERP29 | - |
| - | - | EXOC5 |
| FLNA (filamin) | FLNA | FLNA |
| G3BP2 | - | G3BP2 |
| - | IQGAP1 | IQGAP1 |
| - | KIF5B (kinesin-1) | KIF5B |
| - | MAP1B | - |
| MYH9/10 (myosin II) | MYH9 | MYH9 |
| NAPA/α-SNAP | - | - |
| RAB2A[3] | RAB2A | - |
| - | RAB4A | - |
| RAB7A[3] | RAB7A | - |
| RAB11A[3] | - | - |
| RAB14[3] | - | - |
| - | SEC22B (SNARE) | - |
| - | SEC24C (COPII) | - |
| TMED10 (COPI) | - | - |
| VCP | - | - |

[1] [32]

[2] [33]

[3] colocalization with SMM shown by fluorescence microscopy in [32]

The interaction of STM with the early secretory system is poorly characterized. In fact, the involvement of early secretory host factors, e.g. RAB2A, in SMM biology was only recently described by proteomic studies [32, 33]. We now expand this interaction by showing the physiological relevance of RAB1A in SIF formation (Fig 4) and presence of both RAB1A and RAB1B on SIF (Fig 5). The direct association of RAB1A/B with SIF possibly connects several distinct trafficking events. First, RAB1B was shown to be involved in formation of the COPI vesicle coat, which participates in *intra*-Golgi and retrograde Golgi-to-ER transport [89–91]. Second, the COPI components COPA and COPG1 were shown to partly colocalize with SIF [32]. In accordance, our screen identified the majority of COPI components as mid- or high-ranking hits (ARCN1, COPA/B1/B2/G1, Table 1 and Fig 3). Thus, RAB1A and/or RAB1B might represent a physical link between COPI vesicles and SCV and/or SIF for the redirection of early secretory material as depicted in Fig 8iv.

The physical interaction of SIF with COPI vesicles might be, similar to the late endo-/lysosomal fusion machinery, additionally accompanied by tethering factors and SNAREs. The conserved oligomeric Golgi (COG) tethering complex was shown to be a RAB1 effector and directly bind COPI components [92, 93]. Interestingly, all components of the COG present in the trafficome scored mid- to high-ranking (COG1/2/3/5/7, S2 Table). Additionally, COG binds STX5, a SNARE that is part of several ER-Golgi and *intra*-Golgi transport-related SNARE complexes [94–97]. Srikingly, only the components of one distinct SNARE complex, comprising STX5, GOSR2/GS27/membrin, BET1, and SEC22B, scored all mid- to high-ranking (Table 1 and S2 Table). STM effectors partaking in this interaction might by SseF and SseG, as they were recently shown in the recent BioID screen [88] to interact with STX5 and SEC22B, besides PipB2 also interacting with SEC22B. However, whether the potential involvements indicated by these collective data expand the potential RAB1/COPI interaction cascade described above remain to be elucidated.

Another tethering factor, the transport protein particle (TRAPP) complexes were identified as RAB1 guanine exchange factors (GEFs) [98–101] and COPI tethers [TRAPPII; 100, 102]. TRAPPI is the core shared by all TRAPP complexes, with II and III possessing unique additional subunits. TRAPPC8, the unique component of TRAPPIII, scored high-ranking. Other components were not present in this trafficome, except the TRAPP core subunit TRAPPC2, which unexpectedly did not score at all (S2 Table). So far TRAPPIII is only characterized to participate in autophagy [101, 103, 104]. Besides, the tethering golgin USO1/p115 scored mid-ranking (S2 Table). USO1 is also a RAB1B effector and COPI tether [91, 105], executing these roles partly in conjunction with COG [106]. Additionally, USO1 is able to bind STX5 [107]. As for the COG complex and ER/Golgi SNAREs, for both, the TRAPP complexes and USO1, a specific role in SIF biogenesis remains to be elucidated.

It has already been described that STM, depending on SseF/SseG, recruits exocytic vesicles from the Golgi apparatus destined to the plasma membrane to the SCV [108]. Which host factors are involved in this process was unclear, and our work now sheds light on this phenotype by showing the presence of exocytic RABs, i.e. RAB3A, RAB8A, and RAB8B (Fig 5), and exocytic SNAREs, i.e. VAMP2, VAMP3, and VAMP4 (Fig 6), on SIF.

Besides their involvement in exocytosis, VAMP4 and VAMP3 are also known to prominently participate in endosome-to-Golgi transport in conjunction with STX16, VTI1A, and STX6 or STX10 for EEs or LEs, respectively [109, 110].This might represent another interaction cascade of SIF as VTI1A was a mid-ranking hit (although STX6 and STX10 were not included in the trafficome and STX16 ranked low, S2 Table). Moreover, the STM-mediated redirection of LAMP1-containing vesicles from the Golgi apparatus to the early SCV was shown to involve recruitment of STX6 and VAMP2 via SPI1-T3SS effector SipC [69]. Alternatively, this might happen via the SPI2-T3SS effector PipB2 that was identified in the recent

BioID screen as an interactor of VAMP2 [88]. Furthermore, in homotypic EE fusion STX16 is replaced by STX13 [111], and STX13 was previously shown to be present on early SCV [74, 112]. While the exact role of RAB3A and the identity of SNAREs involved in the processes described above remain to be determined, this might indicate that the interception of secretory vesicles by SMM depends on a SNARE complex comprising a distinct combination of the abovementioned SNAREs as represented in Fig 8ii.

In addition to its exocytic role, the mid-ranking hit RAB8A is involved in recycling processes as indicated by the localization on tubular recycling endosomes (RE) [113]. This localization depends on several factors such as the RAB8 GEF RAB3IP/RABIN8 (which is also part of the trafficome, though it ranked low, S2 Table), concurrently being an effector of the RE master regulator RAB11 [114]. Another factor is the RE-localized MICALL1, which interacts with the dynamin-like ATPases EHD1 and EHD3 [113, 115, 116]. Interestingly, MICALL1 was identified in an RNAi screen with focus on STM intracellular replication [31], EHD1 and EHD2 were present in the proteome of maturing SCV [33], and EHD4 was present in late SMM [32]. Moreover, the association of the maturing SCV and late SMM with RAB11A/B was shown previously [32, 50], with RAB11A following RAB1A as the second-highest-ranking RAB in our screen (S2 Table). The SPI2-T3SS effector SopD2 most likely plays a role in RAB8 recruitment, as it was previously shown to interact with RAB8 [88, 117]. Collectively, these data strongly argue for a continued association of STM not only with exocytic compartments as described above involving a distinct SNARE complex, but also with recycling compartments with RAB8 isoforms at its center at later time points (summarized in Fig 8ii).

Data on the involvement of clathrin-coated structures or adaptor protein complexes in STM pathobiology are scarce. We now show an association of clathrin via one of its light chains, CLTA, with SMM (Fig 7). It is peculiar that two proteomic studies [32, 33], as well as our screen, indicate an involvement of the AP-2 complex in biogenesis of SMM, while the other adaptor complexes were not identified. The presence of the CME-related AP-2 is remarkable as it is primarily plasma membrane-localized, in contrast to the Golgi traffic-related AP-1 and AP-4, or the endo-/lysosomal traffic-related AP-3 and AP-5 [43]. Especially AP-3 deserves detailed analyses in the future since its two core subunit isoforms scored in mid- and high-ranking range (AP3B1 and AP3D1, S2 Table, see Fig 8i). Several SPI2-T3SS effectors, i.e. PipB2, SopD2, and SseG, might participate in such a recruitment because the recent BioID screen revealed the interaction with various AP-2 and AP-3 core subunits [88]. However, the examination of other AP complexes also seems worthwhile, since the latter study indicates interactions with several of them and the trafficome screen did not comprehensively cover AP complexes.

In summary, we successfully employed a sub-genomic RNAi screen to systematically identify new host factors, corresponding protein complexes, and pathways involved in SIF formation. By providing physiologically relevant data regarding SIF formation, this work further corroborates involvements of host factors with SMM indicated by previous proteomics studies [32, 33]. Similar future screens can also reveal the biogenesis of several other SIT [15], and extend to the host cell types important for *Salmonella* pathogenesis.

## Materials and methods

### Bacterial strains and growth conditions

For infection STM NCTC 12023 WT and isogenic SPI2-T3SS-defective strain P2D6 harboring plasmid pFPV-mCherry/2 or isogenic GFP-expressing MvP1897 were used (for details see S3 Table). Strains were routinely grown in Luria-Bertani (LB) broth (Difco, BD, Heidelberg, Germany) containing 50 μg/mL carbenicillin for plasmid selection at 37˚C with aeration.

## Cell lines and cell culture

Experiments were performed using the parental HeLa cell line (ATCC No. CCL-2) or the lenti-virus-transfected HeLa cell line stably expressing LAMP1-GFP [18]. Cells were routinely cultured in Dulbecco's modified Eagle's medium (DMEM) containing 4.5 g/L glucose, 4 mM stable glutamine, and sodium pyruvate (Biochrom, Berlin, Germany) supplemented with 10% inactivated fetal calf serum (iFCS; Gibco, Darmstadt, Germany) in an atmosphere of 5% $CO_2$ and 90% humidity at 37˚C.

## siRNA library and individual siRNAs

The siRNAs used here were part of a human whole-genome library obtained from Qiagen (Hilden, Germany) deposited at the Max Planck Institute (MPI) for Infection Biology (Berlin, Germany). The actual siRNA library is a custom-made library similarly built as others from the MPI [118, 119] and comprised siRNAs targeting 496 host proteins with a threefold coverage, i.e. three individual siRNAs per target. The targets are mostly involved in intracellular trafficking as they were all chosen from GO terms associated with trafficking except the terms 'autophagy' and 'canonical glycolysis' (see S1 Table for a full list of the parental GO terms). A volume of 4 μL of each siRNA (0.2 μM, end concentration of 5.2 nM) was spotted automatically onto 96-well Clear Bottom Black Cell Culture Microplates (Corning, Corning, NY, USA) and frozen at -20˚C before transfer. The 1,488 individual siRNAs were distributed on 24 96-well plates in total per biological replicate with three biological replicates performed ($n$ = 3). Additionally, each plate contained the same amount of the following siRNAs from Qiagen as knockdown controls: AllStars as negative and Hs_PLK1_7 as positive controls. A custom siRNA from Qiagen directed against SKIP served as a phenotype-specific control [22] and was spotted on location. Information including target sequences for these siRNAs and those ordered for validation experiments, are listed in S4 Table.

## Reverse transfection with siRNA

If not using 96-well screening plates as detailed above, the amount for an end concentration of 5 nM siRNA was spotted onto standard cell culture 6-well plates (for mRNA extraction or Western blot analyses; TPP, Trasadingen, Switzerland) or 8-well polymer bottom chamber slides (for quantification of SIF formation; μ-Slides, ibidi, Martinsried, Germany).

Next, a mixture of the transfection reagent HiPerFect (Qiagen, Hilden, Germany) and serum-free cell culture medium was applied and this was incubated for 5–10 min at room temperature (RT). Subsequently, 5,000, 125,000, or 20,000 cells per well of 96-well plates, 6-well plates, or 8-well chamber slides, respectively, were added in serum-containing medium and incubated for 72 h at 37˚C in a humidified atmosphere containing 5% $CO_2$.

## Gene expression quantification

After reverse transfection with siRNAs in 6-well plates, total RNA of cells was extracted using the RNeasy Mini Kit following the manufacturer's instructions (Qiagen, Hilden, Germany). Homogenization during extraction was performed using Qiagen QIAshredder columns. Then, 1 μg of RNA digested with DNaseI (NEB, Frankfurt a. M., Germany) was used for reverse transcription of mRNA with the RevertAid First Strand cDNA Synthesis Kit (Thermo Scientific, Dreieich, Germany) following the manufacturer's instructions employing the Oligo(dT)$_{18}$ primer. For RT-PCR, 1 μL of cDNA was used with the Thermo Scientific Maxima SYBR Green/Fluorescein qPCR Master Mix (2x). As reference gene, the housekeeping gene *GAPDH* was selected [120]. For control of individual host factor knockdowns, primers were used

employing the PrimerBank database [121, 122]. Primers for the host factors analyzed and control *GAPDH* are listed in S5 Table. Primer concentration was 150 nM each, and primer efficiency was determined for each primer pair. RT-PCR was performed in an iCycler instrument (Bio-Rad, Munich, Germany) in triplicates in 96-well plates. Relative expression was determined using the $2^{-\Delta Ct}$ method [84, 123] with *GAPDH* expression set as 100%. Results were plotted using SigmaPlot 11 (Systat Software, Erkrath, Germany).

## Western blot analyses

Whole cell lysates were prepared using a lysis buffer (1% Triton X-100, 5% glycerol in phosphate-buffered saline [PBS] with cOmplete, EDTA-free Protease Inhibitor Cocktail; Roche, Mannheim, Germany) from reversely transfected cells in 6-well plates. The resulting extracts were centrifuged at 1,800 x *g* and the supernatant was quantified for protein content with the Pierce BCA Protein Assay Kit (Thermo Scientific) using BSA as standard following the manufacturer's instructions. After precipitation by addition of a five-fold volume of acetone and incubation for 1 h at 4˚C, pellets were dried and resuspended in SDS-PAGE loading buffer. Of precipitated whole cell protein, 30 μg was loaded onto 10% (for SKIP and VPS11) or 12% (RAB7A) gels and separated by SDS-PAGE. After electrophoresis, samples were blotted onto a 0.45 μm nitrocellulose membrane using a semi-dry electrophoretic transfer unit (Bio-Rad). Blots were incubated with primary antibodies directed against SKIP (dilution 1:1,000; custom), VPS11 (dilution 1:1,000; sc-100893, Santa Cruz Biotechnology, Dallas, TX, USA), or RAB7A (dilution 1:1,000; #9367/clone D95F2, Cell Signaling Technology, Frankfurt a. M., Germany), or γ-tubulin (dilution 1:1,000; T6557/clone GTU-88, Sigma-Aldrich, Taufkirchen, Germany) as loading control. Secondary antibodies coupled to horseradish peroxidase were chosen according to the donor species of the primary antibodies and diluted 1:10,000. Detection was achieved by an ECL detection kit (Thermo Scientific), and blots were visualised with a ChemiDoc imaging system (Bio-Rad). Densitometric analysis was performed with ImageLab (v4.0, Bio-Rad).

## Construction of plasmids

Plasmids used in this study were either obtained from Addgene, kind gifts from various laboratories, or cloned by Gibson Assembly or restriction enzyme digests and are listed in S3 Table. Oligonucleotides for the construction of plasmids encoding host proteins fused to mRuby2 or EGFP are listed in S5 Table. First, N- or C-terminal mRuby2 vectors were cloned. For that, the vectors pEGFP-C1 and pEGFP-N1 were amplified and EGFP was exchanged for a fragment encoding mRuby2. Genes encoding host proteins were amplified from vectors obtained from DNASU (S3 Table) and then inserted into mRuby2 vectors by Gibson Assembly. Plasmids encoding host proteins fused to EGFP were constructed using restriction enzyme digests. The vector pEGFP-C3 was digested with *Kpn*I and *Xba*I or *Kpn*I and *Bam*HI and the larger fragment was recovered. The inserts were treated the same way and fragments were ligated.

## Host cell transfection

For LCI for the localization of host factors, HeLa or HeLa LAMP1-GFP cells were seeded 1 d prior to transfection. About 20,000 or 150,000 cells were seeded in 8-well chamber slides (see above) or 3.5 mm glass bottom dishes (FluoroDish, WPI, Berlin, Germany), respectively. For transfection 0.5 or 2 μg of plasmid DNA in 25 or 200 μL serum-free medium were mixed with 1 or 4 μL of FuGENE HD transfection reagent (DNA to reagent ratio of 1:2; Promega, Mannheim, Germany) and incubated for 10 min at RT. Medium on the cells was changed and the

transfection mixture applied. Cells were incubated for at least 18 h before infection with medium change during infection. For a complete list of transfection plasmids, see S3 Table.

### Infection experiments

Overnight cultures of STM were diluted 1:31 and grown for additional 3.5 h in LB broth in glass test tubes with agitation in a roller drum at 60 rpm. HeLa cells were infected with STM WT or *ssaV* serving as control for screening approaches in 96-well plates with a multiplicity of infection (MOI) of 15 ($OD_{600}$ of subcultures ranged from 3.3–4.2 and 3.0–4.3 for WT or *ssaV*, respectively), otherwise for colocalization analysis or SIF quantification in 8-well chamber slides or FluoroDishes with an MOI of 75 or 50, respectively. Infection only of 96-well plates was synchronized by centrifugation at 500 x g for 5 min, and in all cases proceeded for 25 min at 37˚C in a humidified atmosphere containing 5% $CO_2$. Cells were washed thrice with full medium or PBS for screening or non-screen LCI purposes, respectively, and incubated in full medium containing 100 µg/mL gentamicin for 1 h to eliminate extracellular bacteria. Then medium containing 10 µg/mL gentamicin was applied for the remainder of the experiment.

### Live cell imaging

For LCI full medium was replaced by imaging medium consisting of Minimal Essential Medium (MEM) with Earle's salts, but without $NaHCO_3$, L-glutamine and phenol red (Biochrom, Berlin, Germany) and instead supplemented with 30 mM HEPES (4-(2-hydroxyethyl)-1-piperazineethanesulfonic acid) (Sigma-Aldrich), pH 7.4, containing 10 µg/mL gentamicin. Fluorescence imaging for screening purposes was performed using a Zeiss Cell Observer microscope with Yokogawa Spinning Disk Unit CSU-X1 (Carl Zeiss, Göttingen, Germany), Evolve 512 x 512 EMCCD camera (Photometrics, Tucson, AZ, USA), automated PZ-2000 stage (Applied Scientific Instrumentation, Eugene, OR, USA), and infrared-based focus system Definite Focus, operated by Zeiss ZEN 2012 software (blue edition). The microscope was equipped with live cell periphery consisting of a custom-made incubation chamber surrounding the microscope body and connected with "The Cube" heating unit (Life Imaging Services, Basel, Switzerland) maintaining 37˚C and the Incubation System S for $CO_2$ and humidity supply (PeCon, Erbach, Germany). Images were acquired using the Zeiss LD Plan-Neofluar 40x/ 0.6 Corr air objective (with bottom thickness correction ring). For acquisition of GFP and mCherry BP 525/50 (Zeiss) and LP 580 (Olympus, Hamburg, Germany) filters, respectively, were applied. Imaging of individual screening plates was executed hourly from 1–7 h p.i. with eight positions per well and a single Z-plane per position (adjusted to the plane with the majority of bacteria). All images obtained were processed by the ZEN software. Non-screen LCI was performed using a Leica SP5 confocal laser-scanning microscope (CLSM) operated by Leica LAS AF software. The microscope was also equipped with live cell periphery consisting of 'The Box' incubation chamber (Life Imaging Services, Basel, Switzerland), a custom-made heating unit and a gas supply unit 'The Brick'. Images were acquired using the HCX PL APO CS 100x/ 1.4 oil objective (Leica, Wetzlar, Germany), applying the polychroic mirror TD 488/543/633 for acquisition of GFP and mCherry. All images were processed by LAS AF software.

### Quantification of SIF formation

After siRNA knockdown and infection as described above, 100 infected HeLa LAMP1-GFP cells per condition as indicated were examined live with a 40x objective from 6–8 h p.i. for the presence of SIF as exhibited by WT-infected cells, and the percentage in relation to siAllstars-treated WT-infected cells calculated. Results from three independent experiments (*n* = 3) were plotted using SigmaPlot 11.

## Data analysis

For the central entry and collection of scoring data, the MATLAB-based utility SifScreen was used. The categorization of targets/hits was executed using the GO classification scheme [124, 125]. For the visualization of protein interactions, the STRING v10 database with default settings was applied [126].

## Supporting information

**S1 Text. Considerations for screen design and setup.**
(DOCX)

**S1 Table. Full list of the 496 host factors targeted in the siRNA screen including gene symbols, NCBI Gene IDs and corresponding accession numbers, UniProt entry no. and corresponding entry names, official full names, and aliases.**
(XLSX)

**S2 Table. Summary of the analysis of the executed trafficome siRNA screen with lists of the scoring of all targets, the scoring of the hits only, and the scoring of low-, mid-, and high-ranking hits (scoring of 1–4, 5–7, or ≥8, respectively; see main text for scoring details) for comparison.**
(XLSX)

**S3 Table. Bacterial strains and plasmids used in this study.**
(DOCX)

**S4 Table. Individual siRNA information used for validation.**
(DOCX)

**S5 Table. Oligonucleotides used in this study.**
(DOCX)

**S1 Fig. Representative fields of view for siRNA-silenced and STM-infected cells at late time points of the screen.** HeLa cells expressing LAMP1-GFP (green) were reverse transfected with the indicated siRNAs for 72 h (also corresponding to Fig 4). Then, cells were infected with STM WT expressing mCherry (red) at MOI = 15, and imaged by SDCM. Depicted are representative field of views 7 h p.i. Scale bar, 20 μm.
(TIF)

**S2 Fig. Validation of host factor siRNA silencing.** HeLa LAMP1-GFP cells were left untreated or reverse transfected with siAllStars or the indicated siRNA. A) For RT-PCR, total RNA was extracted, mRNA reverse transcribed, and the generated cDNA was used in RT-PCR. Depicted are means and standard deviation for three biological replicates ($n = 3$), each performed in triplicates. Statistical analysis was performed against siAllstars with Student's $t$-test and indicated as: ***, $p < 0.001$. B) For Western blot analysis, cell lysates were processed to determine the protein levels of SKIP, RAB7A, and VPS11. Two independent knockdown assays are indicated by k/d 1 and k/d 2. As a loading control, blots were additionally processed for detection of γ-tubulin. C) Densitometry of Western blot signals for the indicated proteins.
(TIF)

**S1 Movie. Time-lapse imaging of siAllStars-treated infected cells.** The movie corresponds to Fig 1D.
(AVI)

**S2 Movie. Time-lapse imaging of siSKIP-treated infected cells.** The movie corresponds to Fig 1D.
(AVI)

## Acknowledgments

We thank Monika Nietschke and Ursula Krehe for construction of plasmids and technical assistance. Special thanks go to Markus C. Kerr (Brisbane), André P. Mäurer, Peter Braun, Marion Rother, and Thomas F. Meyer (Berlin) for advice and logistics in setting up the screen. We thank Martin Aepfelbacher (Hamburg), Thierry Galli (Paris), Wanjin Hong (Singapore), and Yulong Li (Beijing) for providing transfection vectors and Stéphane Méresse (Marseille) and Christian Ungermann (Osnabrück) for antibodies.

## Author Contributions

**Conceptualization:** Alexander Kehl, Michael Hensel.

**Data curation:** Alexander Kehl, Christopher John.

**Formal analysis:** Alexander Kehl, Vera Göser, Tatjana Reuter, Viktoria Liss, Maximilian Franke, Christopher John.

**Funding acquisition:** Michael Hensel.

**Investigation:** Alexander Kehl, Vera Göser, Tatjana Reuter, Viktoria Liss, Maximilian Franke, Christopher John, Michael Hensel.

**Methodology:** Alexander Kehl, Jörg Deiwick.

**Project administration:** Alexander Kehl, Michael Hensel.

**Resources:** Alexander Kehl, Jörg Deiwick.

**Software:** Christian P. Richter.

**Supervision:** Alexander Kehl, Michael Hensel.

**Validation:** Tatjana Reuter.

**Visualization:** Vera Göser, Tatjana Reuter, Viktoria Liss.

**Writing – original draft:** Alexander Kehl, Michael Hensel.

**Writing – review & editing:** Alexander Kehl, Michael Hensel.

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
