## [Decision Letter · Decision Letter 0]

17 Jan 2020

Dear Prof. Dr. Hensel,

Thank you very much for submitting your manuscript "A trafficome-wide RNAi screen reveals deployment of early and late secretory host proteins and the entire late endo-/lysosomal vesicle fusion machinery by intracellular Salmonella" for consideration at PLOS Pathogens. As with all papers reviewed by the journal, your manuscript was reviewed by members of the editorial board and by several independent reviewers. In light of the reviews (below this email), we would like to invite the resubmission of a significantly-revised version that takes into account the reviewers' comments.

We cannot make any decision about publication until we have seen the revised manuscript and your response to the reviewers' comments. Your revised manuscript is also likely to be sent to reviewers for further evaluation.

Sincerely,

Guy Tran Van Nhieu

Section Editor

PLOS Pathogens

Guy Tran Van Nhieu

Section Editor

PLOS Pathogens

Kasturi Haldar

Editor-in-Chief

PLOS Pathogens

orcid.org/0000-0001-5065-158X

Michael Malim

Editor-in-Chief

PLOS Pathogens

orcid.org/0000-0002-7699-2064

Reviewer's Responses to Questions

**Part I - Summary**

Reviewer #1: The intracellular pathogen Salmonella enterica is characterized by the formation of the Salmonella-containing vacuole (SCV) and a unique network of various Salmonella-induced tubules (SIT). The bacteria effector proteins required for the formation of SCV and SIF are well-defined; in contrast, the corresponding host proteins are poorly characterized.

In this study the authors aim to identify new host factors responsible for for the formation of SCV and SIF using functional RNAi screen. They identify four different classes of host proteins: i) the late endo-/lysosomal SNARE; ii) proteins involved in the early secretory pathway; iii) proteins in the late secretory pathway, and iv) proteins involved in clathrin-coated structures. Although the study generates a large amount of data, the majority of the study is descriptive and provides limited mechanistic insights into the formation of SCV and SIF. Since the study provides only minor improvement over previous studies, such as PMID 24274083，22701604， 25348832，26084942, I feel that multiple major issues should be addressed before the manuscript is considered for publication.

Reviewer #2: The study is essentially a screen of the effect of the knockdown of host protein on the

formation of Salmonella-containing vacuole (SCV and Salmonella-induced filaments (SIF). 496 genes already known to be involved in protein trafficking were selected to be knocked-down by siRNA, and the functional effects were examined by using high-resolution live cell imaging to score effects on SIF induction, dynamics and morphology. Several proteins were identified to have functions in SIF formation, including the late endo-/lysosomal SNARE complex proteins ( STX7, STX8, VTI1B, and VAMP7/ VAMP8), RAB7, RAB1A and RAB1B, RAB3A, RAB8A, RAB8B,VAMP2, VAMP3, VAMP4 and clathrin chains. The validation experiments included microscopy experiments, in which the select proteins containing a fluorescent tag were expressed and their interaction with SIF shown by co-localization.

While the study is potentially interesting, there are several issues, which would need to be, in my opinion, adressed prior to the publication.

Major issues:

1. The entire study was performed in HeLa cell line, which is a cervical cancer cell line. Although this cell line has been extensively used, the authors should seek confirmation of the observation in other cell lines, ideally primary cells.

2. The co-localization experiments by using microscopy studies should be accompanied by other types of observations such as western blotting. SCV positioning experiments could also be done, such as in this article: D’Costa, V.M., Coyaud, E., Boddy, K.C. et al. BioID screen of Salmonella type 3 secreted effectors reveals host factors involved in vacuole positioning and stability during infection. Nat Microbiol 4, 2511–2522 (2019) doi:10.1038/s41564-019-0580-9. This article should also be cited and discussed.

At present, the study is largely descriptive and no new function has be claimed for the identified proteins in the SIF formation other than their upstream function.

3. The screen is performed by using imaging system, but the timing of the picture acquisition has not been well described. Since total 496 experiments were performed (plus controls), what was the time from experiment 1 to the last experiment? Could this timing affect the result? Moreover, replicate number for the screen is not clearly explained. In the caption to figure 4 the replica number n=3 is included but it is not mentioned for the main screen. Also, does n=3 mean that three different wells, three different plates or three different times that the experiment was performed?

4. The screen contained the select proteins already previously known to be involved in the vesicular and membrane systems, therefore the results are not surprising and possibly not very novel either. It is still unclear how the library was designed.

5. The authors uses siRNA for the screen and comment e.g. in line that loss of function might not be 100%. The only information whether the siRNA worked well is qPCR data for a subset of these proteins (11 if I counted well?). Further, it is unknown whether the protein levels were affected at the tested time point and concentration, which ultimately dictate the function. For some proteins, these siRNA conditions might need to be optimized (long half-life etc.) or even shRNA might been to be used.

Other minor comments:

1. Line 526: Was the OD600 monitored for the bacterial cultures?

2. Line 560 “Quantification of SIF formation” – the paragraph does not contain sufficient description.

3. Table 1 – the scoring cutoffs should be provided with the reference where to find information about the scoring. Uniprot accession numbers should be introduced (particularly as Uniprot is used as the localization information source) as well as references to the source of information for localization apart from Uniprot.

4. Table 2. The description should be improved. Uniprot on NCBI accession numbers should be referenced.

5. Line 1106: Mention that these are date from the current study.

6. Line 1120-1121: Mention time after transfection

7. Figure 1. Figure D is too small in proportion to Figs. A- C, especially C

8. Figure 2. The information about the time(s) of infection and MOI are needed

9. Figure 4. How many cells were counted per image?

10. Figure 5. It is left unclear why in certain cases cells were co-transfected with LAMP1 by using transiently expressing LAMP1 and in other cases stable transfection was chosen.

11. Table S2 and S3. Improve the description of the table (Caption) to increase the clarity.

12. Line 1199: What is special plastic vs standard plastic? Unclear.

13. Appears that there are only 11 targets for q PCR but over 400 genes we initially knocked down.

14. Figure 2. how long did the phenotypic screening take? was that considered while interpreting the changes in time?

15. More images could be made available for the protein targets, whose knockdown affects the SIF formation. The attached tables are not very informative.

Reviewer #3: In this manuscript, a time-resolved high-content siRNA screen was performed to identify host factors involved in the dynamics of “Salmonella induced filaments”, membrane structures that are pertinent during enterocyte infection of S.Tm. The focus was on genes involved in cellular trafficking. Therefore, it has not been too surprising that a large number of the tested siRNAs showed some alterations in the Sifs. Nevertheless, the authors were able to extract very interesting data and novel links through their screen. They show the involvement of a specific SNARE complex involved in endo-lysosomal trafficking, novel roles of RabGTPases, new links with the recycling and secretion machinery and unexpected involvement of clathrin structures. The study represents an important research effort and the results are useful to the scientific community. The controls of the screen could be described in a clearer way, also showing the limitations of such screens. Showing a bit more functional analysis for one of the four proposed research directions (through the four hit groups) either through the usage of bacterial mutants, or through the application of inhibitors, or biochemical follow up would provide an important boost. Left as it is, the provided study is useful and very much appreciated by this reviewer, however it would not make the most of the performed screen.

**Part II – Major Issues: Key Experiments Required for Acceptance**

Reviewer #1: 1． Since the formation of SCV and SIF is relatively latter events in the Salmonella invasion, some host factors, such those related with clathrin-coated structures, could alter the formation of SCV and SIF via impacting bacterial invasion．How do the authors distinguish host factors directly impacting the formation of SCV and SIF from the indirect modulators?

２.　Colocalization cannot be used as evidence of direct or indirect interaction between two proteins (line 239). Thus, discussions related with Figures 4-7 don’t make sense to me.

３.　HGS was chosen as it is the highest-ranking hit among the screen. However, knockdown of HGS did not reduce SIF formation. What is the explanation?

Reviewer #2: 1. The entire study was performed in HeLa cell line, which is a cervical cancer cell line. Although this cell line has been extensively used, the authors should seek confirmation of the observation in other cell lines, ideally primary cells.

2. The co-localization experiments by using microscopy studies should be accompanied by other types of observations such as wester blotting. SCV positioning experiments could also be done, such as in this article: D’Costa, V.M., Coyaud, E., Boddy, K.C. et al. BioID screen of Salmonella type 3 secreted effectors reveals host factors involved in vacuole positioning and stability during infection. Nat Microbiol 4, 2511–2522 (2019) doi:10.1038/s41564-019-0580-9. This article should also be cited and discussed.

At present, the study is largely descriptive and no new function has be claimed for the identified proteins in the SIF formation other than their upstream function.

3. The screen is performed by using imaging system, but the timing of the picture acquisition has not been well described. Since total 496 experiments were performed (plus controls), what was the time from experiment 1 to the last experiment? Could this timing affect the result? Moreover, replicate number for the screen is not clearly explained. In the caption to the figure 4 the replica number n=3 is included but it is not mentioned for the main screen. Also, does n=3 mean that three different wells, three different plates or three different times that the experiment was performed?

Reviewer #3: Considering the analysis pipeline, it is not entirely clear ihow the authors assessed the impact on invasiveness. For example, lower than 50% phenotypes (that would be hits) could also be caused by loss of invasiveness or cell death.

The majority of candidate genes showed a certain degree of interference, therefore one wonders whether this does reflect the involvement of intracellular trafficking or is there the possibility of low specificity? The authors could provide controls with another small collection of siRNA irrelevant to endocytic trafficking.

A number of candidates have been implicated, and described in S.Tm. intracellular trafficking. Therefore, it would be great to have a bit more follow up- do they act in the same pathway? The authors provide 4 directions, however with very little functional follow up or confirmation through alternative approaches, such as inhibitors. Another good addition would be a link with S.Tm effectors, for example the authors could test this through a verification of SseF and SseG’s role in the recruitment of screened factors.

Through the hits, the authors propose a diverse origin of host membranes for the Sifs, however this lacks solid evidence. Therefore, it should be down-toned a bit.

It would help if the authors could be more structured about the points they want to make in the discussion. It appeared a bit as a mini review

**Part III – Minor Issues: Editorial and Data Presentation Modifications**

Reviewer #1: 1. The microscopy images shown are often very small, making it difficult to determine subcellular localization.

2. Please check grammar throughout the manuscript.

Reviewer #2: Minor points/editing:

1. The article should be re-written in a more clear manner. There are issues throughout with the sentence clarity, for instance the authors often uses undefined pronouns in the beginning of the sentence, which makes it uneasy for the reader to follow the meaning (e.g. sentence in lines 90, 179, 184, 257, 272, and others).

2. Examples of other unclear sentences:

Line 121-122 “a connection to clathrin-coated structures.” Unclear. Connection of what to what?

Line 279-280: “this screen identified AP2A1 as being present on SMM/SCV or scoring high-ranking (Table S2), respectively” unclear.

283-284: “Accordingly, the main coat determinants implicated in the formation of CCVs were among the high-scoring hits, including both clathrin light chains” What does authors mean by coat determinants?

Line 311: “Proteomics shown presence or absence of host factors on the organelle of

Interest (…)”. I would rephrase this to be “Proteomic analyses lead to the…”

and cite the literature.

Line 315-316 “However, a functional role revealed by RNAi does not necessarily require colocalization of the host factor with the compartment, because a function

may be mediated indirectly, involving several interacting partners. “ what function? Which compartment? The sentence is overall unclear.

Line 360-361: “It is also a RAB1B effector and COPI and COPII tether [97, 115, 116],

361 partly in conjunction with COG [117], besides being likewise able to bind STX5 [118]. “ rephrase this sentence. Non grammatical.

Line 426-427: “It is peculiar that proteomics, as well as our screen, indicate an involvement of the AP-2 complex, but one of the other adaptor complexes. This is noteworthy…” Both sentence are unclear. First sentence does not specify which proteomics experiment is referred to, and it is also unclear what the involvement of AP-2 might be – in what? The second part of the sentence is not grammatical. The second sentence starts with the undefined pronoun, which is highly discouraged and introduces lack of clarity.

3. The abstract, Instead of providing points, maybe make could be re-written in a more exciting way.

4. The supplementary text:” Considerations for screen design and setup” should be rewritten in a more concise and clear manner. Also, pay attention to the use of undefined pronouns.

Reviewer #3: Undefined abbreviations: LCI, AAA, VCP etc.

Choice of wordings:

Wording line 226: “exactly quantified”

Wording line 294: “out data”

PLOS authors have the option to publish the peer review history of their article (what does this mean?). If published, this will include your full peer review and any attached files.

Reviewer #1: No

Reviewer #2: No

Reviewer #3: No
---

## [Decision Letter · Decision Letter 1]

19 May 2020

Dear Prof. Dr. Hensel,

We are pleased to inform you that your manuscript 'A trafficome-wide RNAi screen reveals deployment of early and late secretory host proteins and the entire late endo-/lysosomal vesicle fusion machinery by intracellular Salmonella' has been provisionally accepted for publication in PLOS Pathogens.

Best regards,

Guy Tran Van Nhieu

Section Editor

PLOS Pathogens

Guy Tran Van Nhieu

Section Editor

PLOS Pathogens

Kasturi Haldar

Editor-in-Chief

PLOS Pathogens

orcid.org/0000-0001-5065-158X

Michael Malim

Editor-in-Chief

PLOS Pathogens

orcid.org/0000-0002-7699-2064

Reviewer Comments (if any, and for reference):

Reviewer's Responses to Questions

**Part I - Summary**

Reviewer #1: The authors have addressed the points that I have raised previously. Although there are additional room to further improve the provided study, such as utilizing methods complementary to the high-content siRNA screen, it will be unrealistic to perform (and unwise to request) these experiments considering the current coronavirus situation in the authors' city and throughout the world.

Reviewer #2: The authors made effort to address the concerns. The only outstanding issue is the fact that the study has been performed in HeLa cells and not at least validated in other, physiologically relevant cell line or primary cells. I will leave it up to the Editor to decide whether this type of confirmatory study is necessary for the publication in Plos Pathogens.

**Part II – Major Issues: Key Experiments Required for Acceptance**

Reviewer #1: (No Response)

Reviewer #2: (No Response)

**Part III – Minor Issues: Editorial and Data Presentation Modifications**

Reviewer #1: (No Response)

Reviewer #2: (No Response)

PLOS authors have the option to publish the peer review history of their article (what does this mean?). If published, this will include your full peer review and any attached files.

Reviewer #1: No

Reviewer #2: No

---

## [Editor Report · Acceptance letter]

24 Jun 2020

Dear Prof. Dr. Hensel,

We are delighted to inform you that your manuscript, "A trafficome-wide RNAi screen reveals deployment of early and late secretory host proteins and the entire late endo-/lysosomal vesicle fusion machinery by intracellular Salmonella," has been formally accepted for publication in PLOS Pathogens.

Best regards,

Kasturi Haldar

Editor-in-Chief

PLOS Pathogens

orcid.org/0000-0001-5065-158X

Michael Malim

Editor-in-Chief

PLOS Pathogens

orcid.org/0000-0002-7699-2064